# A New Wind Speed Forecasting Modeling Strategy Using Two-Stage Decomposition, Feature Selection and DAWNN

**Sizhou Sun** [1,2,*] **, Lisheng Wei** [1] **, Jie Xu** [1] **and Zhenni Jin** [1]

1  College of Electrical Engineering, Anhui Polytechnic University, Wuhu 241000, China; lshwei_11@163.com (L.W.); xujie_1288@126.com (J.X.); jinzhenni@163.com (Z.J.)
2  School of Mechatronic Engineering and Automation, Shanghai University, Shanghai 200072, China
*  Correspondence: sszhou12345@i.shu.edu.cn; Tel.: +86-138-5532-3560

**Abstract:** Accurate wind speed prediction plays a crucial role on the routine operational management of wind farms. However, the irregular characteristics of wind speed time series makes it hard to predict accurately. This study develops a novel forecasting strategy for multi-step wind speed forecasting (WSF) and illustrates its effectiveness. During the WSF process, a two-stage signal decomposition method combining ensemble empirical mode decomposition (EEMD) and variational mode decomposition (VMD) is exploited to decompose the empirical wind speed data. The EEMD algorithm is firstly employed to disassemble wind speed data into several intrinsic mode function (IMFs) and one residual (Res). The highest frequency component, IMF1, obtained by EEMD is further disassembled into different modes by the VMD algorithm. Then, feature selection is applied to eliminate the illusive components in the input-matrix predetermined by partial autocorrelation function (PACF) and the parameters in the proposed wavelet neural network (WNN) model are optimized for improving the forecasting performance, which are realized by hybrid backtracking search optimization algorithm (HBSA) integrating binary-valued BSA (BBSA) with real-valued BSA (RBSA), simultaneously. Combinations of Morlet function and Mexican hat function by weighted coefficient are constructed as activation functions for WNN, namely DAWNN, to enhance its regression performance. In the end, the final WSF values are obtained by assembling the prediction results of each decomposed components. Two sets of actual wind speed data are applied to evaluate and analyze the proposed forecasting strategy. Forecasting results, comparisons, and analysis illustrate that the proposed EEMD/VMD-HSBA-DAWNN is an effective model when employed in multi-step WSF.

**Keywords:** two-stage signal decomposition; hybrid backtracking search optimization algorithm; wavelet neural network; wind speed forecasting

## 1. Introduction

Sustainability transitions are beneficial transformation processes that make social production and consumption sustainable through socio-technical systems. Major power structural changes from the fossil-fuel energy systems to low-carbon and renewable energy supply are one vital aspect of the sustainability transitions [1,2]. The large-scale green renewable low-carbon wind resources that support the sustainable development of energy systems has attracted worldwide attention. The exploration and utilizations of wind power have large potentials to reduce large greenhouse gas emissions and promote sustainable economic development. However, large-scale integration of wind power into power grid greatly influences the reliability, safety, and power quality of the power system for the

stochastic and fluctuate characteristics of wind speed [3]. As wind speed is one important factor for wind power generation, precise WSF can eliminate influences, therefore, accurate WSF techniques are critical issues for the wind farms [4].

Over the past decade, more reliable and accurate WSF approaches have been developed to solve these problems of the inherent volatility in wind speed time series. There are three main categories of models for short-term WSF, i.e., statistical approaches, artificial intelligent (AI) methods, and hybrid forecasting models.

Persistence method [5], autoregressive moving average (ARMA) [6], autoregressive integrated moving average (ARIMA) model [4,7], Kalman filter (KF) [8], and other statistical approaches make WSF by constructing a relationship between the historical data samples and the predicted values [9]. Back-propagation neural network (BPNN) [10], Elman neural networks (ELMNN) [11], support vector machine (SVM) [12], wavelet neural network (WNN) [13], radial basis function neural network (RBFNN) [14], multilayer perceptron (MLP) [15,16], and least square support vector machine (LSSVM) [17] are typical AI approaches that have been widely employed in short-term WSF. The wavelet neural network (WNN) approach can model the complex non-linear relationships among wind speed data through regression and training, thus, WNN is adopted as the prediction engine in the proposed forecasting strategy.

The random and irregular nature of wind speed make the AI approaches more suitable for WSF than statistical approaches. However, these pure forecasting models suffer from large errors regardless of statistical models or AI approaches when they make WSF directly without pre-processing wind speed data. To enhance the forecasting accuracy, the hybrid methods taking advantages of individual merits have drawn widespread attentions for time series-based WSF. Hu et al. [17] employed decomposition-based forecasting approach combining empirical wavelet transform (EWT) with LSSVM tuned by coupled simulated annealing (CSA). Meng et al. [18] presented a new hybrid model using wavelet packet decomposition (WPD) and BPNN tuned by crisscross optimization algorithm for multi-step WSF. Wang et al. [19] examined the forecasting performance of a hybrid WSF model using WNN tuned by genetic algorithm (GA) and variational mode decomposition (VMD). These hybrid forecasting models firstly utilize single signal decomposition technique to break the original empirical wind speed into different sub-series, then, the sub-series variables are employed as the inputs of AI methods for WSF. To avoid excessive manual intervention and improve adaptivity, intelligent optimization algorithms are applied to tune the parameters combination of AI methods. Seen from the experimental results in the previous literatures, hybrid forecasting models appear to be obviously more precise as compared with single forecasting models.

The previous literatures illustrate that AI models combined with multi-scale decomposition techniques achieve satisfactory forecasting results, however, single signal decomposition methods cannot often thoroughly tackle the non-stationary and non-linear components in the wind speed. Thus, there still exists much room for improvement in wind speed pre-processing [20]. Sun et al. [21] proposed an improved EEMD with a BPNN model for short-term WSF. The hybrid EEMD-SVM model proposed by Hu et al. [12] discarded the highest frequency IMF1 directly to reduce the influence of the most disorder and highest frequency components on the forecasting accuracy. Yu et al. [11] developed three hybrid models, namely EMD-singular spectrum analysis (SSA)-ELMNN, EEMD-SSA-ELMNN, and CEEMDAN-SSA-ELMNN, for 1 h average WSF. In these forecasting models, the SSA approach is applied to extract the trend of the most irregular component IMF1. The retreatment of IMF1 by SSA method can significantly improve the forecasting performance. Peng et al. [22] proposed a novel hybrid model combining AdaBoost-ELM with two-stage decomposition approach for multi-step ahead WSF. In the two-stage decomposition process, the complementary ensemble empirical mode decomposition with adaptive noise (CEEMDAN) is firstly used to break the empirical wind speed data into IMFs and one Res, then, the most nonstationary IMF1 is further decomposed by VMD technique. In a similar way, Yin et al. [23] presented an effective secondary decomposition technique to eliminate the nonstationary characteristics in the samples. The forecasting results of different Cases show that

the hybrid models with two-stage decomposition or secondary decomposition technique can yield higher accuracy.

As stated by [24,25], not all input candidates promote the final forecasting results positively and there exists some illusive components generated by the decomposition technique or measurement error. Energy measurement and kernel density estimation-based Kullback–Leibler divergence are exploited by Jiang et al. [26] as a feature selection technique to identify the effective candidates so as to eliminate negative influence from the illusive components. Salcedo-Sanz et al. [24] developed coral reefs optimization as a feature selection technique to select the effective variables from the total empirical samples for reducing the input dimension number. Apart from the feature selection technique and parameter optimization method alone for the forecasting engine to enhance the prediction accuracy, these functions can be realized simultaneously through the hybrid optimization algorithm. For example, a hybrid gravitational search algorithm (HGSA) integrating real-value GSA (RGSA) with binary-value GSA (BGSA) is exploited by Luo et al. [27] to undertake feature selection and parameter optimization, thereinto, exploiting the BGSA to identify the vital feature variables from the total compound feature selection, and the RGSA algorithm is applied to tune the parameter combination in ELM.

Inspired by the forecasting mechanism in the previous literatures, a novel combined strategy using two-stage decomposition technique, hybrid backtracking search optimization algorithm, WNN with double activations through weighted coefficient (DAWNN), namely EEMD/VMD- HBSA-DAWNN, is proposed for short-term WSF. The proposed model is trained and tested using two sets of randomly selected wind speed data from a wind farm of Anhui in China. With respect to similar studies in WSF, the main work and contributions of this article are illustrated as follows.

- The proposed forecasting strategy takes advantage of individual methods, including two-stage decomposition technique, HBSA, DAWNN, that can enhance forecasting accuracy. The proposal not only thoroughly tackles the nonstationary characteristics of wind speed data, but also remedies the deficiencies of the AI approach.
- To decompose thoroughly, a two-stage decomposition technique combining EEMD with VMD is exploited to deal with wind speed data, eliminating the characteristic of irregularities.
- To better improve the regression performance of WNN, double activation functions combining the Morlet function with the Mexican hat function by weighted coefficients, namely DAWNN, are proposed.
- As stated by Zheng et al. [28], there are ineffective candidate features in wind speed time series that generate negative influence on forecasting results, and WNN with inappropriate parameters tends to get stuck in over-fitting or under-fitting easily. The parameters of WNN should be tuned for the improvement of forecasting accuracy. Thus, the BBSA algorithm is applied to remove ineffective candidate variables, while RBSA is used to tune the input weights, output weights, and weighted coefficients in DAWNN other than random selection for the proposed forecasting engine. The feature selection technique and parameter optimization are realized by HBSA algorithm.

The remaining parts of this article are arranged as follows. In Section 2, the detailed WSF strategy are introduced. In Section 3, two-stage wind speed decomposition technique, HBSA algorithm and DAWNN are presented. The statistical error evaluation indices and the procedure of model construction and development are presented in the Section 4. Numerical results and discussion are illustrated in Section 5. In the end, the conclusions are drawn. The abbreviations of technical terms are listed in Appendix A.

## 2. The Proposed WSF Strategy

The overall working process of the proposed WSF model depicts in Figure 1. The proposed hybrid model EEMD/VMD-HBSA-DAWNN makes WSF in following steps.

- **Step 1:** In this step, the original empirical wind speeds are decomposed into several modes, different IMFs and a Res using two-stage decomposition technique EEMD/VMD to eliminate the irregular and fluctuant characteristics of wind speed for better forecasting. Firstly, the original empirical samples are broken into different IMF*s* and one Res, then, the IMF1 generated by EEMD is further decomposed by VMD approach into several modes.

- **Step 2:** Considering the positive influences of normalization on the performance of WNN, the input variables are linearly normalized into interval [0, 1]. Prior to WSF using the proposed DAWNN method, partial autocorrelation function (PACF) values are applied to determine the correlation coefficients among the input candidates for establishing the training and testing input-matrix.

- **Step 3:** As illustrated in *Algorithm 1*, select the effective candidates through feature selection by BBSA technique and optimize the parameters combination of DAWNN model by RBSA algorithm using the decomposed wind speed data, which are realized by HBSA simultaneously.

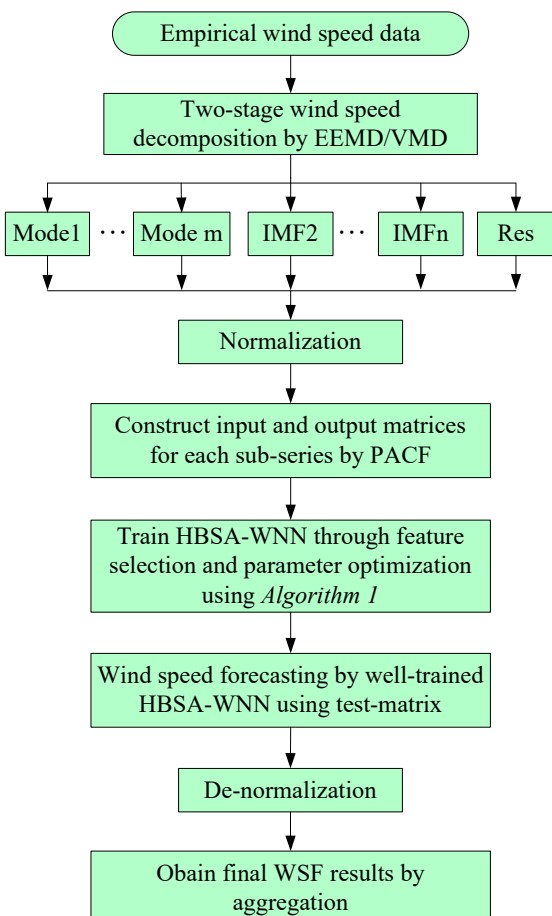

**Figure 1.** Overall framework of EEMD/VMD-HBSA-DAWNN model.

## 3. Theoretical Background

### 3.1. Two-Stage Wind Speed Decomposition Technique

#### 3.1.1. EEMD

A novel noise-assisted signal processing technology EEMD was developed by Wu et. al [29] for eliminating the mode mixing shortage of EMD. As stated by Jiang [26], EEMD is an effective adaptive signals analysis approach which decomposes and analyzes nonlinear signals by an iterative sifting process. In each sifting process, a finite amplitude Gaussian white noise is added into the decomposed

signal and the white noise can be removed through averaging. For wind speed samples $x(t)$, the main signal analysis process of EEMD is illustrated as follows.

- **Step 1:** A Gaussian white noise is added into the empirical wind speed data, and new signal is generated as Equation (1),

$$X(t) = x(t) + n(t),\tag{1}$$

 where $n(t)$ stands for the Gaussian signal.

- **Step 2:** Disassemble the new signal into $N$ different amplitude-frequency IMFs and a Res by EMD algorithm, then, the $X(t)$ can be expressed as Equation (2),

$$X(t) = \sum_{i=1}^{N} IMF_i + \text{Res},\tag{2}$$

 where N stands for the decomposed quantity.

- **Step 3:** Repeat Step 1 and **2** by adding different Gaussian white noise at each sifting process.
- **Step 4:** Eliminate the Gaussian white noise and obtain the final IMFs by averaging all the corresponding IMFs in the end.

### 3.1.2. VMD

A novel signal analysis approach VMD is proposed by Dragomiretskiy and Zosso [30] in 2014 to break down real-value signals into different models with specific sparsity properties. The decomposition process is carried out through solving the minimum value optimization problem, which is expressed in Equation (3),

$$\min\left\{\sum_k||\partial_t[(\delta(t) + \tfrac{j}{\pi t}) \otimes u_k(t)]e^{-j\omega_k t}||_2^2\right\}$$
$$s.t.\textstyle\sum_k u_k = f(t)\tag{3}$$

where $f(t)$ and $u_k$ stand for the original signal and its $k$th component, respectively. $\delta(t)$, $\otimes$, $\omega_k$ and $k$ denote the Dirac distribution, convolution operator, the center frequency of $u_k$, and the number of modes, respectively.

To simplify the above constraint optimization problem, Lagrangian multipliers and quadratic penalty term are utilized to construct Lagrangian function expressed in Equation (4),

$$\ell(\{u_k\}, \{\omega_k\}, \lambda) := \alpha \sum_{k=1}^{K} ||\partial_t[(\delta(t) + \tfrac{j}{\pi t}) \otimes u_k(t)]e^{-j\omega_k t}||_2^2 + ||f(t) - \sum_{k=1}^{K} u_k(t)||_2^2 + \langle \lambda(t), f(t) - \sum_{k=1}^{K} u_k(t) \rangle,\tag{4}$$

where $\alpha$ and $\lambda$ stand for the balancing value of the data-fidelity constraint and Lagrange multiplier, respectively.

To address the optimization, the Alternate Direction Method of Multiplier (ADMM) was developed to obtain the saddle point of the augmented Lagrangian function during the iterative sub-optimization. As results, the solutions of Equation (4) can be obtained by Equations (5) and (6):

$$\hat{u}_k^{n+1}(\omega) = \frac{\hat{u}(\omega) - \sum_{i\neq k}\hat{u}_i(\omega) + \frac{\hat{\lambda}(\omega)}{2}}{1 + 2\alpha(\omega - \omega_k)^2},\tag{5}$$

$$\omega_k^{n+1} = \frac{\int_0^{\infty} \omega|\hat{u}_k(\omega)|^2 d\omega}{\int_0^{\infty} |\hat{u}_k(\omega)|^2 d\omega}.\tag{6}$$

where $\hat{u}_k^{n+1}(\omega)$, $\hat{u}(\omega)$, $\hat{u}_i(\omega)$ and $\hat{\lambda}(\omega)$ are the Fourier transforms of $u_k^{n+1}(\omega)$, $u(\omega)$, $u_i(\omega)$ and $\lambda(\omega)$, respectively.

*3.2. The Working Principle of HBSA*

3.2.1. RBSA

BSA, proposed by Civicioglu [31], is applied to determine the optimal solution of the objective function, which is used to improve the performance of system through the input information and optimization condition. In the optimization process, BSA exhibits good exploitation capabilities and robust exploration to find the best values of the population using mutation and crossover operators. BSA has only one parameter that needs to be tuned, therefore, it has been widely applied to deal with the real-valued optimization [32,33]. The optimization process of BSA is generally summarized as the following steps.

- **Step 1:** Initialization

Not only the current population but also the historical population of BSA are generated randomly with uniform distribution according to Equations (7) and (8).

$$P_{i,j} \sim U(low_j, up_j), \tag{7}$$

$$oldP_{i,j} \sim U(low_j, up_j), \tag{8}$$

where $P_{i,j}$ and $oldP_{i,j}$ are current population and history population, respectively. $low_j$ and $up_j$ are the minimal and maximal variables, respectively. $U$ stands for uniform distribution.

- **Step 2:** Selection-I

According to Equation (9), the historical population $oldP$ is redefined through comparing two random variables *a* and *b*. Then, the orders of the individuals are changed through a random shuffle function expressed as Equation (10). $oldP$ remembers a randomly selected previous generation to produce a new trial population by utilizing the prior experience to determine the search direction matrix.

$$oldP = \begin{cases} P & if(a < b|a, b \sim U(0,1)) \\ oldp & otherwise \end{cases}, \tag{9}$$

$$oldP := permuting(oldP), \tag{10}$$

where a and b are random variables within [0, 1].

- **Step 3:** Mutation

BSA's mutation mechanism is mathematically expressed as Equation (11). As seen from the formula, BSA utilizes the information of the historical population to create a mutation population and the previous generation penetrate the whole mutation process. The parameter *F* in Equation (11) is used to adjust the amplitude of the search direction matrix (*oldP-P*),

$$\begin{cases} M = P + F \cdot (old - P) \\ F = 3 \cdot rndn \end{cases}, \tag{11}$$

where *rndn~U(0,1)*.

- **Step 4:** Crossover

Although BSA is improved and developed on basis of differential evolution (DE), the crossover process of BSA algorithm is quite different from DE algorithm. Two predefined strategies are used to determine the matrix map of BSA. One strategy utilizes the parameter *mixrate* to generate a binary integer-valued matrix, as shown in Equation (12), which guides the crossover direction.

$$
\begin{aligned}
&if(rand1 < rand2) \quad map(i, u(1 : mixrate * rand * D)) \\
&else \quad map(i, randi(D)) \quad = 0
\end{aligned}
\tag{12}
$$

where *rand*1 and *rand*2 are random values, D stands for the population dimension.

The second strategy updates the trial population *T*. If $map_{i,j} = 1$, $T_{i,j}$ is replaced by $P_{i,j}$, thus, the final crossover results are shown in Equation (13),

$$
T = map \cdot *P + (\sim map) \cdot *M,
\tag{13}
$$

where *map* stands for binary integer-valued matrix.

- **Step 5:** Selection-II

After calculation of the fitness values of each individual, the comparisons between the fitness values of the original population and the trial population are carried out, and $T_i$ with better fitness values will update $P_i$. Step 2-5 are repeated until the terminal conditions are satisfied. In the end, the population *P* with the best solution is determined and denoted as $P_{\text{best}}$. Here, the proposed DAWNN with two weighted activation functions is trained and tested, i.e., the free parameters and the weighted coefficients of DAWNN are tuned by RBSA, the output errors of DAWNN are utilized as the training objective function of RBSA.

3.2.2. BBSA

RBSA is utilized to solve the real-valued optimization problems in the continuous domain, however, there are many binary-valued discrete parameter problems that need to be dealt with. To solve these binary-valued problems, Ahmed developed a BBSA technique [33]. The BBSA is applied in the similar way to convert the position to 0 or 1 by sigmoid function, as shown in Equations (14) and (15).

$$
S_{i,j} = \frac{1}{1 + e^{-w}},
\tag{14}
$$

$$
B_{i,j} = \left\{
\begin{array}{ll}
0 & S_{i,j} < 0.5 \\
1 & S_{i,j} \geq 0.5
\end{array}
\right. ,
\tag{15}
$$

where $S_{i,j}$, $\omega$ and $B_{i,j}$ is the sigmoid function, population value and binary value, respectively. $B_{i,j}$ is 1 when the $S_i$ value is bigger than or equal to 0.5, otherwise the $B_{i,j}$ is 0. In the WSF model, the input variables of the proposed DAWNN are encoded as binary-valued vector, 1 means "selected" while 0 stands for "discard".

*3.3. WNN*

Wavelet Transform (WT) has been widely utilized in wind speed/power prediction [34]. One popular method is to employ WT as a preprocessor to decompose the historical wind speed or power data into different relatively stable sub-series which are used as input variables of intelligent methods for further regression [13,34]. Wavelet functions are used in the hidden layer of ANN model to construct WNN, which is another popular application of WT [13,35]. The basic structure of a WNN are shown in Figure 2, which contains one input layer, one hidden layer and one output layer. The algebraic representations of the WNN is expressed as Equation (16).

$$
\left\{
\begin{array}{l}
y = \sum\limits_{j=1}^{n} \omega_j F_j \\
F_j = \sum\limits_{i=1}^{m} \psi_{a_{i,j} b_{i,j}} \left( \frac{x_i - b_{i,j}}{a_{i,j}} \right)
\end{array}
\right. ,
\tag{16}
$$

where $a_{i,j}$ and $b_{i,j}$ stands for scale factor and position factor, respectively.

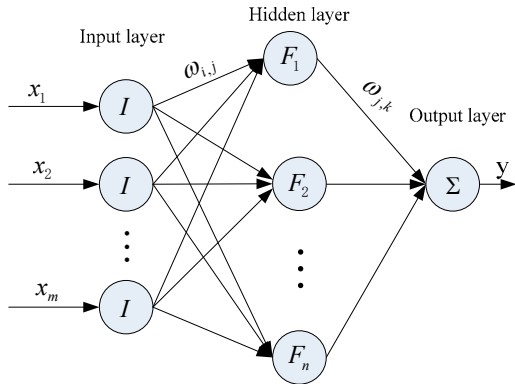

**Figure 2.** Architecture of general wavelet neural network. (*i*, *j* and *k* stand for the number of the input nodes, hidden nodes and output nodes).

Morlet and Mexican hat functions, expressed as Equations (17) and (18), respectively, are typical wavelet functions used in the WNN method, which can be explained by Figure 3. For instance, in Ref. [35], WNN with Mexican hat wavelet activation function combining with multi-resolution analysis method was developed for short-term WSF. It is mentioned in Ref. [13] that WNN with Morlet wavelet function can yield better forecasting performance than that with Mexican hat wavelet function. It is obviously seen that Morlet wavelet function has more vanishing mean oscillatory character with diverse oscillations than Mexican hat function, thus, Morlet wavelet function has better capacity in localizing various high frequency elements in the time domain of severely irregular nonlinear signals [13]. Mexican hat function is obtained by second derivative of Gaussian function and Morlet wavelet function is the single frequency sub-sine function under Gaussian envelope. In this study, we take advantages of individuals of Morlet and Mexican hat wavelet functions to construct a hybrid activation function by weighted coefficient for WNN, which is expressed as Equation (19).

$$\psi_1(x) = \cos(1.75x)e^{-0.5x^2}, \tag{17}$$

$$\psi_2(x) = (1 - t^2)e^{-0.5x^2}, \tag{18}$$

$$\psi_3(x) = \mu_1\psi_1(x) + \mu_2\psi_2(x) = \mu_1\cos(1.75x)e^{-0.5x^2} + \mu_2(1 - t^2)e^{-0.5x^2}, \tag{19}$$

where $\mu_1$ and $\mu_2$ are both within [0, 1] and $\mu_1 + \mu_2 = 1$.

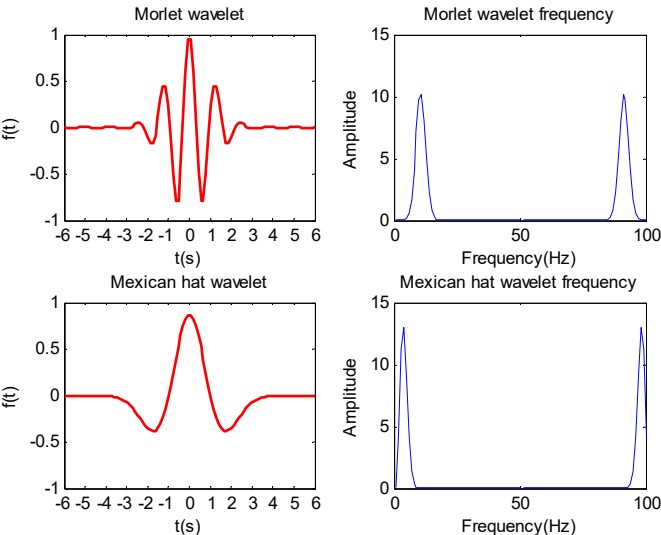

**Figure 3.** Mexican hat and Morlet wavelet functions.

The vector of the real-valued parameter of the proposed DAWNN is expressed as Equation (20). Therefore, there are 5 *n* real-valued values to be tuned by the HBSA.

$$Z = [\omega_{i1}, \omega_{i2}, \cdots, \omega_{in}, \omega_{o1}, \omega_{o2}, \cdots, \omega_{on}, \mu_1, \mu_2, \cdots, \mu_n, a_1, a_2, \cdots, a_n, b_1, b_2, \cdots, b_n], \qquad (20)$$

where $\omega_i$, $\omega_o$ and $\mu$ are the input weighted coefficient, the output weighted coefficient, and weighted coefficient, respectively. *a* and *b* are the same definition as the Equation (16).

The future wind speed value at time step *k* is obtained by WNN through regressive mapping the historical measured wind speed data. In the proposed model, the input vector $[x_1, x_2, \dots, x_m]$ represent the historical wind speed data and *y* is the corresponding forecasting value. The *y* is the output value obtained by a non-linearity autoregressive process expressing in the Equation (21). In addition, the simulated experiments are executed to illustrate the effectiveness of the hybrid activation function.

$$y = x(k + h) = f[x(k), x(k-1), \cdots x(k-n-1)], \qquad (21)$$

where *h* is usually named as the forecasting horizon between the history and the future value, *f* is a nonlinear function describing the relationship between the historical wind speed data and the present value.

### 3.4. The Proposed HBSA-DAWNN Approach

In the HBSA-DAWNN model, the parameter combination in DAWNN are tuned by RBSA method and the effective input variables are selected by BBSA technique. The process of feature selection and parameter optimization for HBSA-DAWNN are presented in the *Algorithm 1*. In the solution representation, feature masks are encoded to binary value 1 or 0. The statistical index root mean square error (RMSE), as expressed in Equation 23, is selected as the fitness function to quantitatively evaluate the HBSA- DAWNN model.

---

**Algorithm 1** The pseudo code of HBSA-DAWNN

---

| | |
|---|---|
| 1: | Generate the initial parameters in BSA: iteration number (*T*), dimension number (*D*), population size (*N*) and *F*. |
| 2: | Set the initial population according to Equation 20. |
| 3: | Convert the population values into binary values according to Equations 14 and 15. |
| 4: | for *i* from *1* to *N* do |
| 5: |   Make WSF by the proposed forecasting engine using each population. |
| 6: |   Calculate the fitness function according to Equation 23 for each population. |
| 7: | endfor |
| 8: | Produce initial historical population according to Equation 8. |
| 9: | for i from 1 to T do |
| 10: |   Update historical population ($oldP_i$) according to Equations 9 and 10. |
| 11: |   Calculate the trial population mutant according to Equation11. |
| 12: |   Convert the population values into binary value according to Equations 14 and 15. |
| 13: |   Determine the input variable matrix using the binary value. |
| 14: |   Calculate the final trial population ($T_{ij}$). |
| 15: |   for i from 1 to N do |
| 16: |     Make wind speed forecasting by the proposed forecasting engine. |
| 17: |     Calculate the fitness function using Equation 23 for each population. |
| 18: |   endfor |
| 19: |   Determine the minimal objective function, and find the optimum parameters combination and the effective input variables. |
| 20: | endfor |
| 21: | Obtain the effective input variables matrix and the optimum parameters combination for forecasting. |

---

## 4. Model Construction and Development

To evaluate the proposed forecasting strategy, multi-step WSF experiments using the historical wind speed are carried out. All the tests are executed in MATLAB 2014*a* environment in personal computer with i5, 3.5 GHz CPU and 8 GB RAM under Windows *8* Operating System. Forecasting results illustrate that the proposed EEMD/VMD-HBSA-DAWNN model yields the highest forecasting accuracy with minimum statistical error indices.

### 4.1. The Statistical Error Evaluation Indices

The forecasting performance of a new developed model for WSF are generally evaluated by multiple statistical error indices. Mean absolute error (MAE), RMSE, mean absolute scale error (MASE), and mean absolute percent error (MAPE), expressed as Equations (22)–(25), are typical statistical error indices that evaluate quantitatively the new developed model and the compared models. The forecasting model performs better when the statistical indices are smaller. RMSE reveals the entire deviation between the measured data and the forecasting values, MAE reflects how similar the forecasting values are to the actual measured ones, whereas the unit-free statistical index MAPE is sensitive to the small changes in the forecasting results. Thus, these statistical indices are suited to evaluate the new developed EEMD/VMD-HBSA-DAWNN model.

$$MAE = \frac{1}{N}\sum_{i=1}^{N}|s(i) - \hat{s}(i)|, \tag{22}$$

$$RMSE = \sqrt{\frac{1}{N}\sum_{i=1}^{N}(s(i) - \hat{s}(i))^2}, \tag{23}$$

$$MAPE = \frac{1}{N}\sum_{i=1}^{N}\frac{|s(i) - \hat{s}(i)|}{s(i)} \times 100\%, \tag{24}$$

$$MASE = \frac{1}{N}\sum_{i=1}^{N}\frac{s(i) - \hat{s}(i)}{\frac{1}{N-1}\sum_{i=2}^{N}|s(i) - s(i-1)|}, \tag{25}$$

where $s(i)$ and $\hat{s}(i)$ represent the actual samples and the forecasting value at time $i$, respectively. $N$ stands for the entire quantity of the wind speed data.

### 4.2. Empirical Wind Speed Time Series

The historical wind speed data, measured each 15 min interval, are gathered and stored in the central computer of a wind farm in Anhui of China, to evaluate and test the proposed forecasting strategy. Two sets of 700 successive wind speed samples, as shown in Figure 4, are selected randomly from 2015 as empirical samples. In the empirical samples, the foregoing 600 wind speed data are utilized to train HBSA-DAWNN model while the subsequent 100 wind speed time series are employed to test the proposed model. The statistical descriptions of the empirical data are listed in the Table 1. From the Figure 4 and Table 1, there are high fluctuations within [1.09, 11.54] and [0.75, 12.57] in the wind speed data.

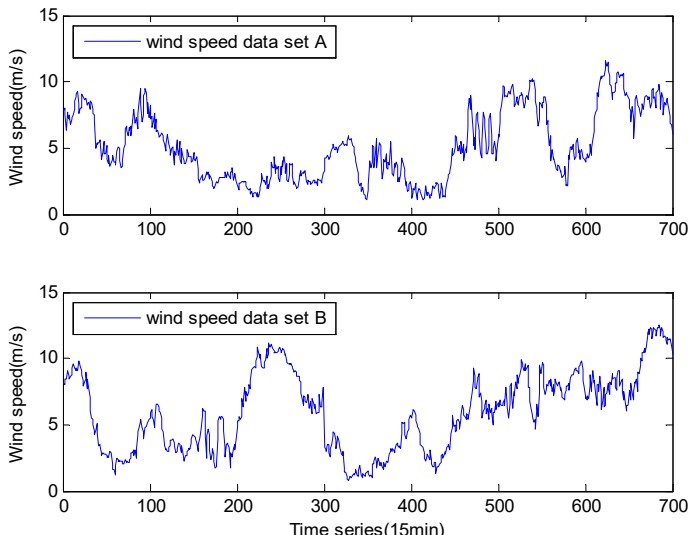

**Figure 4.** Empirical original wind speed data.

**Table 1.** Statistical properties of the total empirical wind speed time series (m/s).

| Data set A | NO. | Max | Median | Min | Mean | St.dev. |
|---|---|---|---|---|---|---|
| Empirical samples | 700 | 11.54 | 4.71 | 1.09 | 5.22 | 2.61 |
| Training data | 600 | 10.19 | 1.22 | 1.09 | 4.66 | 2.31 |
| Test data | 100 | 11.54 | 8.81 | 4.24 | 8.54 | 1.69 |
| **Data set B** | **NO.** | **Max** | **Median** | **Min** | **Mean** | **St.dev.** |
| Empirical samples | 700 | 12.57 | 6.03 | 0.75 | 5.98 | 2.94 |
| Training data | 600 | 11.16 | 0.94 | 0.75 | 5.46 | 2.74 |
| Test data | 100 | 12.57 | 8.45 | 5.99 | 9.14 | 2.01 |

### 4.3. Wind Speed Decomposition Using Two-Stage Decomposition Technique

As proved by previous studies [20,23], further decomposition of IMF1 can improve the forecasting accuracy over horizons of not only one-step but also multi-step prediction while the decompositions of IMF2 and IMF3 make the forecasting accuracy deteriorate, because IMF1 owns the most unsystematic and the highest frequency part of wind speed, which may lower the regression performance of the forecasting engine. To eliminate this negative influence, one practical approach is to directly discard the IMF1 [12]. However, Guo [36] pointed out that this simple treatment can slightly enhance the forecasting performance in some situations. Secondary decomposition method is the other effective approach to solve this problem [23]. Considering EEMD and VMD have good potential to disassemble wind speed into relatively steady sub-series, we attempt to employ it to further deal with the highest frequency IMF1.

The $K$ and $\tau$ in VMD are set 4 and 0, respectively, while the white noise amplitude and the ensemble number in EEMD are set 0.2 and 100. The decomposed results of the empirical samples are shown in Figure 5. From figures, it can be obtained that different IMF$n$ and Res exhibit distinct properties. IMF1 has the highest frequency feature, which reflects random information of the empirical samples. The IMF7, IMF8, and Res reveal their trend characteristics, and the other IMFs reveal periodic characteristics of wind speed.

Figure 5b,d illustrate the decomposed results of IMF1 by the proposed two-stage decomposition technique. The lowest frequency Mode1 exhibits general tendency of IMF1, the highest frequency Mode4 has the smallest contribution to IMF1. After decomposition of wind speed by two-stage decomposition technique, the WSF can be converted into the prediction of each IMF, mode, and Res.

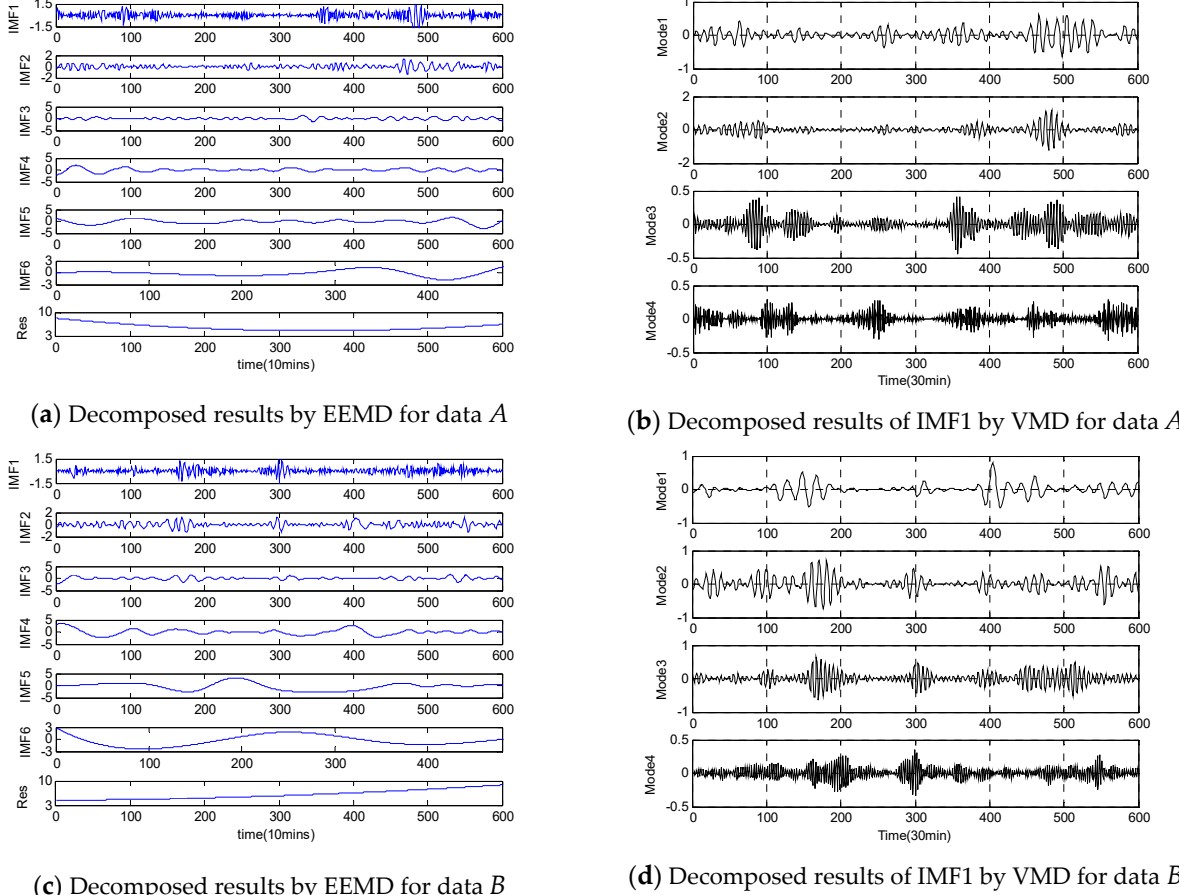

(**a**) Decomposed results by EEMD for data *A*

(**b**) Decomposed results of IMF1 by VMD for data *A*

(**c**) Decomposed results by EEMD for data *B*

(**d**) Decomposed results of IMF1 by VMD for data *B*

**Figure 5.** Wind speed decomposed results by two-stage decomposition technique

*4.4. Construction of Input Feature Matrix for the Forecasting Engine*

Prior to submitting the decomposed components to the propose DAWNN, the input variables matrix should be determined. After linear normalization within [0, 1], PACF technique is applied to determine the input variables combination. The PACF values from lags 0 to 30 are calculated, which are displayed in the Table 2. Seen from the Table 2, PACF value of original wind speed data set *A* is 9. The number 9 reflects the antecedent 9 continuous wind speed contribute mostly to the subsequent forecasting values. As seen from Figure 6, the 9 antecedent wind speed time series are taken as the inputs of the proposed DAWNN to predict the subsequent values. The input combination of the proposed DAWNN for the other time series can be determined in the similar way according to PACF values in the Table 2.

**Table 2.** Lag values obtained by PACF for original wind speed and the different decomposed sub-series.

| Data Set | Original | Mode1 | Mode2 | Mode3 | Mode4 | IMF2 | IMF3 | IMF4 | IMF5 | IMF6 | Res |
|----------|----------|-------|-------|-------|-------|------|------|------|------|------|-----|
| A | 9 | 6 | 5 | 4 | 5 | 12 | 9 | 6 | 8 | 5 | 6 |
| B | 11 | 5 | 6 | 8 | 7 | 10 | 6 | 5 | 7 | 7 | 5 |

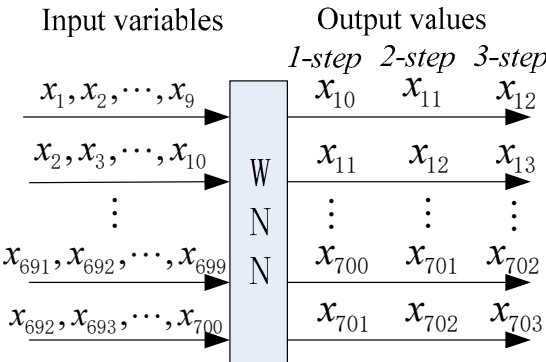

**Figure 6.** Wind speed forecasting format of input variables and output variables for original data A.

### 4.5. Construction of the HBSA-DAWNN

In the BSA-DAWNN model, the dimension of individuals in the population is set according to the sum of $\omega_{i,j}$, $\omega_{j,k}$, $\mu$, $a_j$, and $b_j$. Seen from Figure 3, the outputs of Mexican hat and Morlet wavelet functions are approximately zero when the input values are not in the interval $[-3, 3]$, therefore, the values of $b_{i,j}$ are initialized within $[-3, 3]$. The weighted coefficients $\omega_{i,j}$ and $\omega_{j,k}$ of neural network are generally initialized in the range $[-1, 1]$. For $a_{i,j}$, the interval of $[0.5, 2]$ is selected in that it does not make the wavelet function too spread or dense [13]. All these parameters are initialized with uniform distribution. The parameters in the BSA-DAWNN model are listed in Appendix B.

The number of hidden neurons nodes can also influence the WNN performance. However, there is no clear definition or unified standard with respect to optimal selection of the number of hidden neurons for WNN in different applications. For the original samples A, the quantity of input neurons is set as nine according to Table 2 and the number of the output neuron is set as one. Some experiments are carried out to select the optimal number of the hidden nodes by HAS-WNN model. The statistical index RMSE is shown in Figure 7. From the figure, it can be obtained that the optimal number is 19, where the RMSE is lowest. The optimal hidden node numbers for other ones are selected in the similar manner.

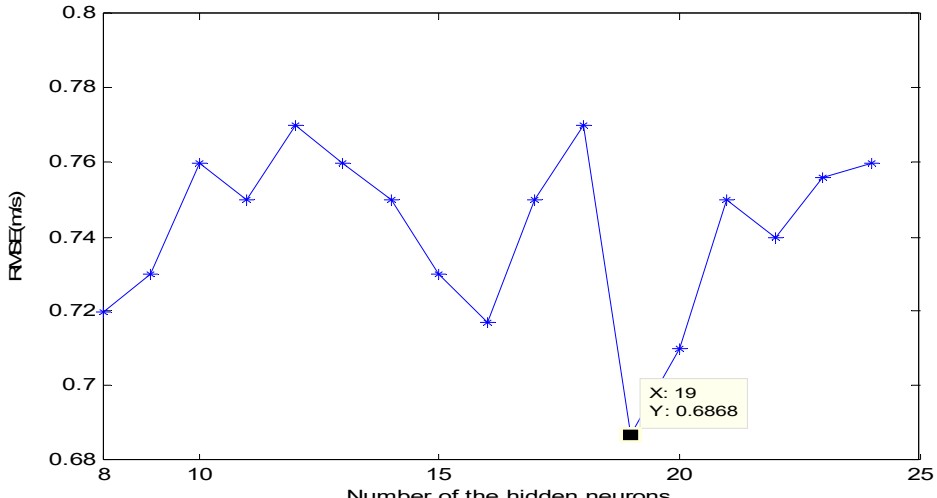

**Figure 7.** The RMSE for different numbers of hidden neurons for original wind data A.

In the HBSA-DAWNN model, the BBSA is exploited to identify the effective candidates from the input matrix predefined by PACF. RBSA is developed to tune the parameters in the proposed DAWNN model. The feature selection results for different time series obtained by BBSA are listed in the Tables 3 and 4.

**Table 3.** Feature selection obtained by HBSA for data set A and its decomposed components.

| Time Series | 1 | 2 | 3 | 4 | 5 | 6 | 7 | 8 | 9 | 10 | 11 | 12 |
|---|---|---|---|---|---|---|---|---|---|---|---|---|
| Original | 1 | 1 | 1 | 0 | 1 | 0 | 1 | 1 | 1 | | | |
| Mode1 | 1 | 1 | 0 | 1 | 1 | 1 | | | | | | |
| Mode2 | 1 | 1 | 1 | 0 | 1 | | | | | | | |
| Mode3 | 1 | 1 | 0 | 1 | | | | | | | | |
| Mode4 | 1 | 1 | 0 | 1 | 1 | | | | | | | |
| IMF2 | 1 | 1 | 0 | 1 | 1 | 0 | 0 | 1 | 1 | 1 | 1 | 1 |
| IMF3 | 1 | 1 | 0 | 1 | 1 | 1 | 0 | 1 | 1 | | | |
| IMF4 | 1 | 0 | 1 | 1 | 1 | 1 | | | | | | |
| IMF5 | 1 | 1 | 0 | 1 | 0 | 1 | 1 | 1 | | | | |
| IMF6 | 1 | 1 | 1 | 1 | 0 | | | | | | | |
| Res | 1 | 1 | 1 | 0 | 0 | 1 | | | | | | |

Note: *1* and *0* mean *selection* and *discard*, respectively.

**Table 4.** Feature selection obtained by HBSA for data set B and its decomposed components.

| Time Series | 1 | 2 | 3 | 4 | 5 | 6 | 7 | 8 | 9 | 10 | 11 |
|---|---|---|---|---|---|---|---|---|---|---|---|
| Original | 1 | 1 | 1 | 0 | 1 | 1 | 1 | 0 | 1 | 0 | 1 |
| Mode1 | 1 | 1 | 1 | 0 | 1 | | | | | | |
| Mode2 | 1 | 0 | 1 | 0 | 1 | 1 | | | | | |
| Mode3 | 1 | 1 | 1 | 1 | 1 | 0 | 1 | 1 | | | |
| Mode4 | 1 | 0 | 0 | 1 | 1 | 1 | 1 | | | | |
| IMF2 | 1 | 1 | 0 | 0 | 1 | 1 | 1 | 1 | 1 | 1 | |
| IMF3 | 1 | 1 | 1 | 0 | 1 | 1 | | | | | |
| IMF4 | 1 | 0 | 1 | 1 | 1 | | | | | | |
| IMF5 | 1 | 0 | 1 | 0 | 1 | 1 | 1 | | | | |
| IMF6 | 1 | 0 | 1 | 1 | 1 | 1 | 1 | | | | |
| Res | 1 | 0 | 1 | 1 | 1 | | | | | | |

Note: *1* and *0* mean *selection* and *discard*, respectively.

## 5. Numerical Results and Discussion

In this study, two categories of prediction models have been constructed and tested: the first category is to develop individual regression models without wind speed decomposition technique, which include Persistence, ARMA, BPNN, and WNN, HBSA-WNN; the second category is to construct signal decomposition-based WSF models, which includes EEMD/VMD-HSBA-DAWNN, EEMD-HSBA-WNN, EEMD/VMD-HSBA-WNN with Morlet function or Mexican hat function, EMD-NN [36] and EEMD-GA-BPNN [37]. The parameters in the EMD-NN and EEMD-GA-BPNN are set according to the respective references. All the forecasting models are tested using the actual wind speed data. As that in a previous study [19], the Persistence model is also utilized as a benchmark approach to evaluate the proposed model.

*5.1. Case 1*

In this case, the empirical wind speed, as displayed in Figure 4a, are employed to construct and evaluate the models. The forecasting results and statistical error indices are presented in Figures 8 and 9, and Tables 5 and 6. As seen from the tables, the smallest error indices are marked in boldface, which means the model exhibits the best forecasting performance. Comparisons are carried out to testify the advantages of the signal decomposition methods, feature selection and parameter optimization. Tables 5 and 6 and Figures 8 and 9 illustrate the following:

- Tables 5 and 6 illustrate that the proposed EEMD/VMD-HSBA-DAWNN model achieves best forecasting results with respect to RMSE with 0.2249 m/s, 0.2681 m/s, and 0.3084 m/s for one-, two-, and three-step, respectively.

- Persistence presents the highest forecasting statistical indices values with 0.8818 m/s, 0.9543 m/s, and 1.0433 m/s RMSE values for one-, two-, and three-step ahead prediction, respectively.
- From Figure 8b,d,f, the accuracy of the individual models is ranked from low to high as Persistence, ARMA, BPNN, WNN and HBSA-WNN.
- According to the statistical index RMSE, Table 6 and Figure 9b,d and f illustrate that the second-highest to the fifth-highest accurate models are EEMD/VMD-HBSA-WNN with Morlet function, EEMD/VMD-HBSA-WNN with Mexican hat function, EEMD-HBSA-DAWNN, EEMD-GA-BPNN, and EMD-NN.
- From the Figures 8 and 9, it is obviously seen that the RMSE values of the WNN-based forecasting strategy are ranked from big to small as WNN, HBSA-WNN, EEMD-HBSA-DAWNN, and EEMD/VMD-HBSA-DAWNN.
- To illustrate the effectiveness of the double activation functions in WNN, the proposed EEMD/VMD-HBSA-DAWNN is compared with the models with Morlet function and Mexican hat function. Tables 5 and 6 illustrate the forecasting results. The RMSE values of the proposed EEMD/VMD-HBSA-DAWNN are the smallest. Thus, the proposed strategy outperforms the model with Mexican hat wavelet function or Morlet function.

**Remark:** *The proposed EEMD/VMD-HSBA-DAWNN outperforms all the other mentioned individual models and hybrid forecasting models when applied in multi-step WSF. The proposed model owns a lower RMSE value 0.2249 m/s whereas RMSEs of 0.7282 m/s, 0.6868 m/s, 0.2631 m/s, 0.2737 m/s, and 0.2599 m/s for the WNN, HBSA-WNN, EEMD/VMD-HBSA-WNN (Morlet), EEMD/VMD-HBSA-WNN (Mexican) and EEMD-HBSA-DAWNN, respectively, in one-step. Likewise, the proposed model produces lower statistical error indices (RMSE, MAE, MAPE, and MASE) in two- and three-step forecasting. The signal decomposition-based hybrid prediction models have better forecasting performance when compared with pure individual forecasting models without signal process.*

- Compared with the single statistical approaches ARMA and Persistence, the single AI models BPNN and WNN yield better forecasting performance in that the artificial intelligent models have better capacities in handling non-linear wind speed time series.
- Signal decomposition-based forecasting models obtain remarkable improvements over the individual forecasting models without signal decomposition methods because there exists non-linearity, non-stable, and high fluctuation in the wind speed time series, the signal techniques break wind speed data into different relatively stationary subseries, thus reducing the regression difficulties of the forecasting engine and improving the prediction accuracy.

- Comparisons between the HSBA-WNN model with Morlet function and the individual WNN model with Morlet function using the same wind speed data are made to examine the capacity of HBSA in feature selection and parameter optimization. From Table 5 and Figure 8, HBSA-WNN performs better than WNN for multi-step prediction. The reasons of these results are that feature selection technique by BBSA method eliminates the illusive components and identify the effective components, while RBSA algorithm is utilized as parameter optimization to tune the parameters in WNN, thus enhancing the forecasting performance.
- The HBSA-DAWNN model with two-stage decomposition method EEMD/VMD has higher forecasting accuracy than the HBSA-DAWNN model with EEMD, whose underlying reasons are that the two-stage decomposition can effectively deal with the problems of the irregularity of IMF1 through further decomposition. Thus, the two-stage decomposition approach EEMD/VMD is an efficient data-preprocessing method in improving WSF performance.
- WNN based on Morlet and Mexican hat activation functions with weighted coefficient outperforms that based on Morlet or Mexican hat activation function, because the proposed

forecasting engine takes advantages of the combination of the individual mother wavelet functions of Morlet function and Mexican hat function by weighted coefficient.

Therefore, the integration of WNN method with feature selection and parameters optimization can provide higher accuracy when applied in WSF. Moreover, the application of two-stage decomposition approach in the wind speed preprocessing is verified by comparing with other models.

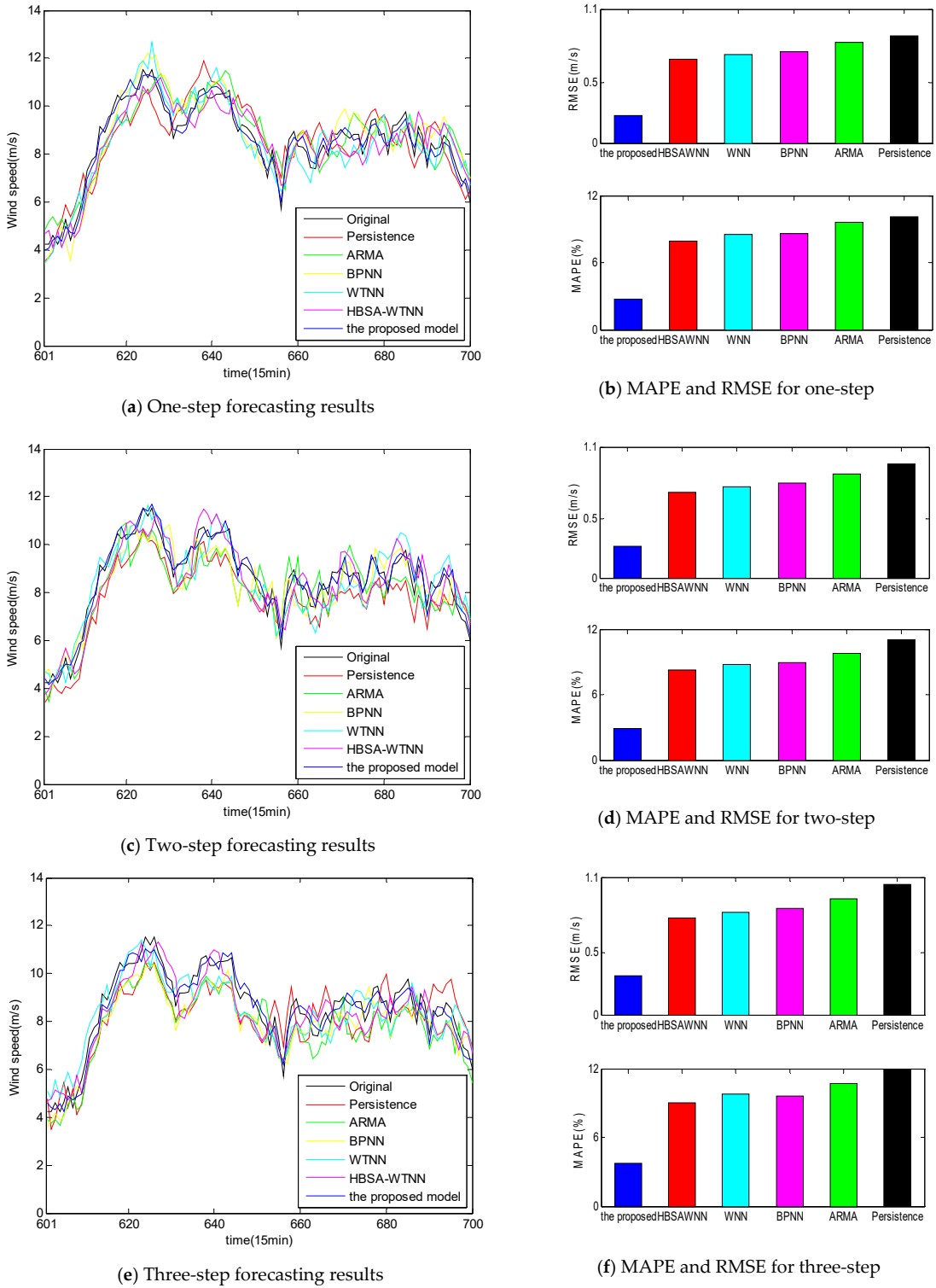

**Figure 8.** Forecasting results by models without signal decomposition for data A.

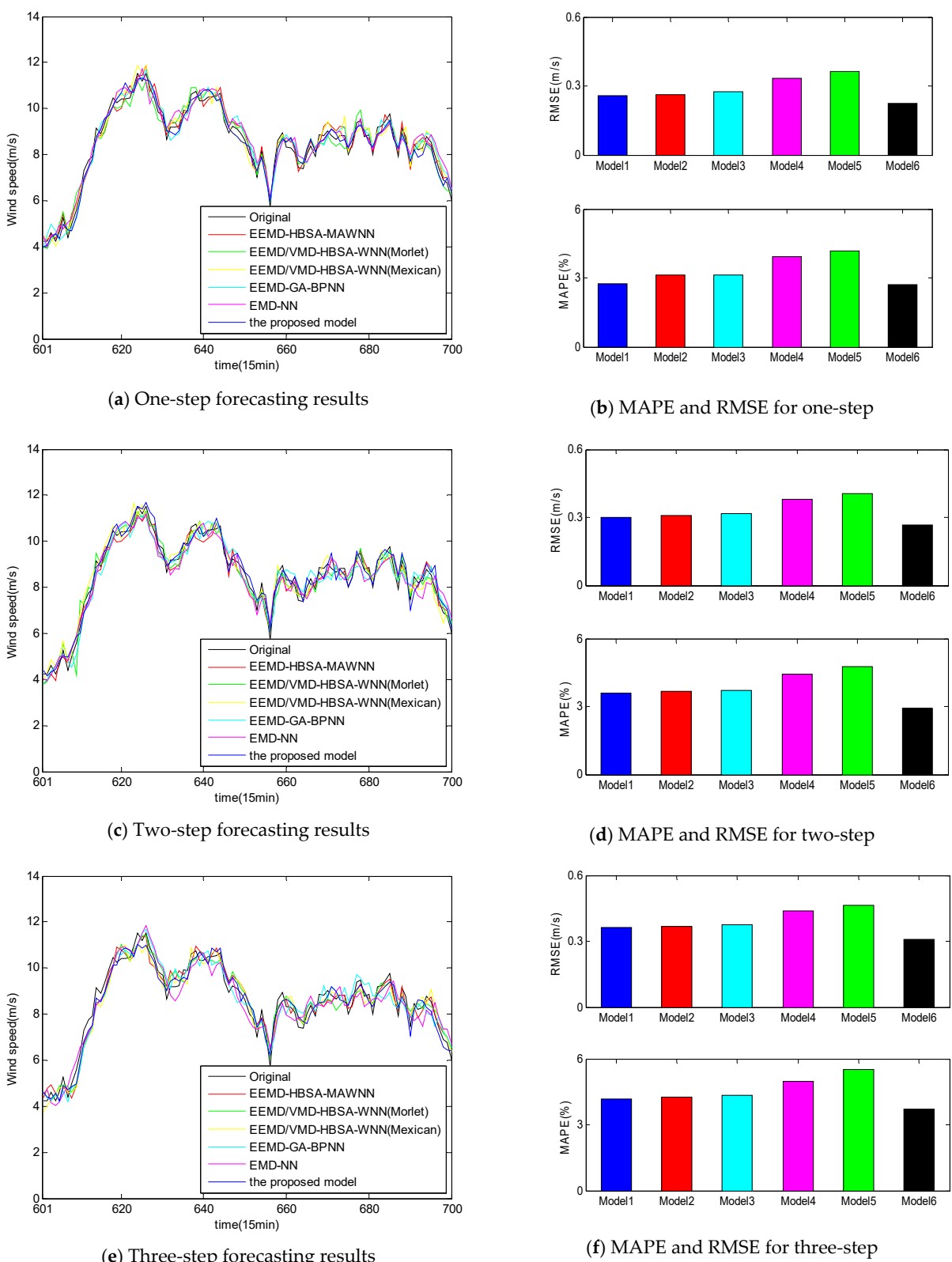

(**a**) One-step forecasting results

(**b**) MAPE and RMSE for one-step

(**c**) Two-step forecasting results

(**d**) MAPE and RMSE for two-step

(**e**) Three-step forecasting results

(**f**) MAPE and RMSE for three-step

**Figure 9.** Forecasting results by models with signal decomposition for data A (Mode1~Model6 are defined as those in Table 6).

**Table 5.** Statistical indices obtained by forecasting models without signal decomposition for data set A.

| Forecasting Horizon | Index | Persistence | ARMA | BPNN | WNN | HBSA- WNN | The Proposed Model |
|---|---|---|---|---|---|---|---|
| One-step | RMSE(m/s) | 0.8818 | 0.8288 | 0.7513 | 0.7282 | 0.6868 | **0.2249** |
| | MAPE (%) | 10.0786 | 9.6457 | 8.5611 | 8.5581 | 7.9595 | **2.6943** |
| | MAE(m/s) | 0.8256 | 0.7789 | 0.6911 | 0.6962 | 0.6473 | **0.2185** |
| | MASE | 1.7097 | 1.6129 | 1.4312 | 1.4417 | 1.3405 | **0.4525** |
| Two-step | RMSE(m/s) | 0.9543 | 0.8719 | 0.7967 | 0.7672 | 0.7228 | **0.2681** |
| | MAPE (%) | 11.0418 | 9.8157 | 8.9655 | 8.7754 | 8.2616 | **2.9087** |
| | MAE(m/s) | 0.9165 | 0.8238 | 0.7487 | 0.7169 | 0.6798 | **0.2311** |
| | MASE | 1.8979 | 1.7061 | 1.5505 | 1.4846 | 1.4078 | **0.4785** |
| Three-step | RMSE(m/s) | 1.0433 | 0.9273 | 0.8501 | 0.8183 | 0.7709 | **0.3084** |
| | MAPE (%) | 12.3091 | 10.7316 | 9.5828 | 9.7785 | 9.0099 | **3.7053** |
| | MAE(m/s) | 1.0178 | 0.8969 | 0.8128 | 0.7935 | 0.7413 | **0.2985** |
| | MASE | 2.1077 | 1.8574 | 1.6833 | 1.6432 | 1.5352 | **0.6182** |

**Table 6.** Statistical indices obtained by forecasting models with signal decomposition for data set A.

| Forecasting Horizon | Index | Model1 [1] | Model2 [2] | Model3 [3] | Model4 [4] | Model5 [5] | Model6 [6] |
|---|---|---|---|---|---|---|---|
| One-step | RMSE(m/s) | 0.2599 | 0.2631 | 0.2737 | 0.3321 | 0.3631 | **0.2249** |
| | MAPE(%) | 2.7453 | 3.1384 | 3.1366 | 3.9048 | 4.1554 | **2.6943** |
| | MAE(m/s) | 0.2328 | 0.2514 | 0.2605 | 0.3119 | 0.3399 | **0.2185** |
| | MASE | 0.4821 | 0.5206 | 0.5394 | 0.6458 | 0.7039 | **0.4525** |
| Two-step | RMSE(m/s) | 0.3013 | 0.3076 | 0.3168 | 0.3784 | 0.4055 | **0.2681** |
| | MAPE(%) | 3.5856 | 3.6612 | 3.7224 | 4.4355 | 4.7689 | **2.9087** |
| | MAE(m/s) | 0.2909 | 0.2861 | 0.3035 | 0.3604 | 0.3883 | **0.2311** |
| | MASE | 0.6023 | 0.5923 | 0.6285 | 0.7463 | 0.8041 | **0.4785** |
| Three-step | RMSE(m/s) | 0.3615 | 0.3651 | 0.3752 | 0.4379 | 0.4647 | **0.3084** |
| | MAPE(%) | 4.1633 | 4.2764 | 4.3314 | 4.9604 | 5.5356 | **3.7053** |
| | MAE(m/s) | 0.3302 | 0.3435 | 0.3537 | 0.4012 | 0.4489 | **0.2985** |
| | MASE | 0.6838 | 0.7113 | 0.7325 | 0.8308 | 0.9296 | **0.6182** |

[1]. Model1 stands for EEMD-HBSA-DAWNN, [2]. Model2 stands for EEMD/VMD-HBSA-WNN with Morlet function, [3]. Model3 stands for EEMD/VMD-HBSA-WNN with Mexican hat function, [4]. Model4 stands for EEMD-GA-BPNN, [5]. Model5 stands for EMD-NN, [6]. Model6 stands for the proposed model.

## 5.2. Case 2

To further manifest the effectiveness of EEMD/VMD-HBSA-DAWNN, Case 2 from the same wind farm is employed to construct the proposed forecasting strategy and other prediction models for multi-step WSF. The empirical wind speed data are shown in the second subgraph of Figure 4. In the same way, the empirical wind speed data are divided into two sets, namely, the first 600 samples are used to train the models and the subsequent 100 ones are employed to test the models. Tables 7 and 8 list th forecasting results, and the corresponding comparisons are carried out. According to the prediction indices in the tables, the same conclusions as that in Case 1 can be obtained.

**Table 7.** Statistical indices obtained by forecasting models without signal decomposition for data set B.

| Forecasting Horizon | Index | Persistence | ARMA | BPNN | WNN | HBSA-WNN | The Proposed Model |
|---|---|---|---|---|---|---|---|
| One-step | RMSE(m/s) | 0.9074 | 0.8379 | 0.7764 | 0.7393 | 0.7097 | **0.2422** |
| | MAPE (%) | 9.8853 | 8.9319 | 8.4734 | 6.8348 | 7.3247 | **2.7088** |
| | MAE(m/s) | 0.8689 | 0.7782 | 0.7349 | 0.6281 | 0.6529 | **0.2355** |
| | MASE | 1.6322 | 1.4618 | 1.3805 | 1.1797 | 1.2265 | **0.4424** |
| Two-step | RMSE(m/s) | 0.9609 | 0.8871 | 0.8201 | 0.7761 | 0.7408 | **0.2771** |
| | MAPE (%) | 10.3226 | 9.7594 | 8.8807 | 8.0361 | 7.0844 | **3.0096** |
| | MAE(m/s) | 0.9151 | 0.8627 | 0.7871 | 0.7213 | 0.6468 | **0.2615** |
| | MASE | 1.7189 | 1.6205 | 1.4783 | 1.3542 | 1.2148 | **0.4913** |
| Three-step | RMSE(m/s) | 1.0581 | 0.9372 | 0.8733 | 0.8312 | 0.7892 | **0.3197** |
| | MAPE (%) | 11.5199 | 9.6726 | 8.7605 | 8.5297 | 8.3009 | **3.2471** |
| | MAE(m/s) | 1.0281 | 0.8717 | 0.7903 | 0.7617 | 0.7406 | **0.2805** |
| | MASE | 1.9311 | 1.6373 | 1.4845 | 1.4307 | 1.3911 | **0.5269** |

**Table 8.** Statistical indices obtained by forecasting models with signal decomposition for data set B.

| Forecasting Horizon | Index | Model1 [1] | Model2 [2] | Model3 [3] | Model4 [4] | Model5 [5] | Model6 [6] |
|---|---|---|---|---|---|---|---|
| One-step | RMSE(m/s) | 0.2887 | 0.2856 | 0.2941 | 0.3495 | 0.3811 | **0.2422** |
| | MAPE (%) | 3.0531 | 3.1792 | 3.2191 | 3.7697 | 4.0994 | **2.7088** |
| | MAE(m/s) | 0.2659 | 0.2742 | 0.2757 | 0.3234 | 0.3581 | **0.2355** |
| | MASE | 0.4996 | 0.5149 | 0.5179 | 0.6074 | 0.6725 | **0.4424** |
| Two-step | RMSE(m/s) | 0.3266 | 0.3203 | 0.3341 | 0.3744 | 0.4105 | **0.2771** |
| | MAPE (%) | 3.3038 | 3.3778 | 3.4375 | 4.0851 | 4.0481 | **3.0096** |
| | MAE(m/s) | 0.2871 | 0.2889 | 0.2967 | 0.3528 | 0.3571 | **0.2615** |
| | MASE | 0.5393 | 0.5426 | 0.5573 | 0.6626 | 0.6706 | **0.4913** |
| Three-step | RMSE(m/s) | 0.3778 | 0.3733 | 0.3897 | 0.4367 | 0.4741 | **0.3197** |
| | MAPE (%) | 3.8426 | 3.7687 | 3.9728 | 4.3296 | 4.7845 | **3.2471** |
| | MAE(m/s) | 0.3321 | 0.3189 | 0.3411 | 0.3864 | 0.4324 | **0.2805** |
| | MASE | 0.6239 | 0.5992 | 0.6405 | 0.7258 | 0.8122 | **0.5269** |

[1]. Model1 stands for EEMD-HBSA-DAWNN, [2]. Model2 stands for EEMD/VMD-HBSA-WNN with Morlet function, [3]. Model3 stands for EEMD/VMD-HBSA-WNN with Mexican hat function, [4]. Model4 stands for EEMD-GA-BPNN, [5]. Model5 stands for EMD-NN, [6]. Model6 stands for the proposed model.

### 5.3. Case 3

The total historical wind speed data of 2015 are displayed in the Figure 10. The red wind speed in the Figure 10a are the selected empirical samples that are used to train and test the proposed model in the previous sections of the paper. To better manifest the effectiveness of EEMD/VMD-HBSA-DAWNN, all the total wind speed data of 2015 were divided into two sets, as shown in Figure 10b,c, to train and test the hybrid models. The first 1st ~ 14640th points wind speed time series are utilized to train the forecasting models to predict the subsequent 2880 wind speed time series. The forecasting curves and their errors of wind speed data A are displayed in the Figure 11 and the statistical indices are listed in Table 9. It can be seen from the Figure 11b,d,f that the forecasting accuracy decreases with the forecasting time. From the Table 9, RMSEs of the proposed model are 0.6516 m/s, 0.7234 m/s and 0.8026 m/s for one-step, two-step and three-step forecasting, respectively, which are smallest compared with EEMD-GA-BPNN and EMD-NN forecasting model. Therefore, the proposed model also outperforms EEMD-GA-BPNN and EMD-NN forecasting models in terms of the statistical indices. From the Table 10, RMSEs of the proposed model are 0.4456 m/s, 0.5018 m/s and 0.5427 m/s for one-step, two-step and three-step forecasting, respectively, which lower than that of EEMD-GA-BPNN and EMD-NN model. Thus, the same conclusions can be drawn for the wind speed data B.

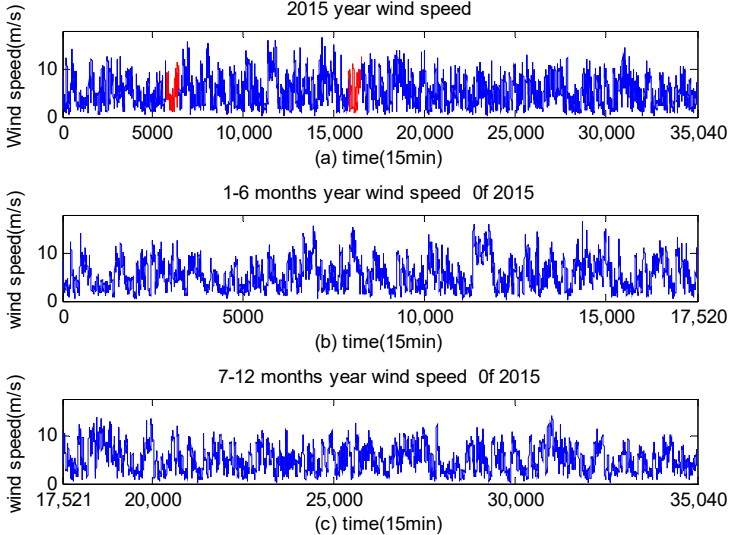

**Figure 10.** Half year and whole year wind speed data of 2015.

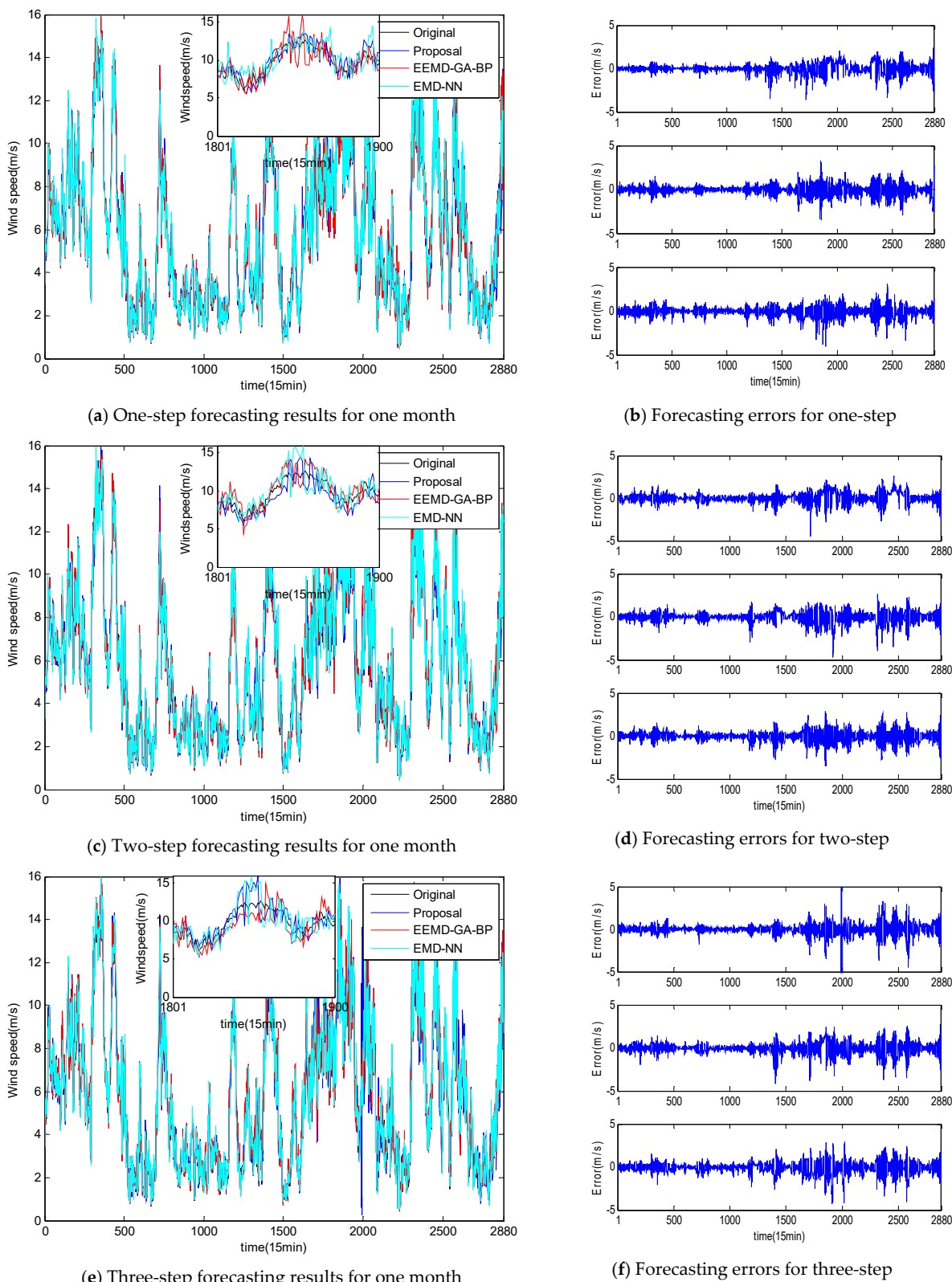

(**a**) One-step forecasting results for one month       (**b**) Forecasting errors for one-step

(**c**) Two-step forecasting results for one month       (**d**) Forecasting errors for two-step

(**e**) Three-step forecasting results for one month       (**f**) Forecasting errors for three-step

**Figure 11.** Wind speed forecasting results for data set A.

**Table 9.** Statistical indices obtained by forecasting models for data set A.

| Horizon | Index | Proposal | EEMD-GA-BPNN | EMD-NN |
|---------|-------|----------|--------------|--------|
| One-step | RMSE(m/s) | 0.6516 | 0.7398 | 0.7634 |
| | MAPE (%) | 8.3655 | 9.2497 | 9.4184 |
| | MAE(m/s) | 0.4721 | 0.5691 | 0.5793 |
| | MASE | 1.0642 | 1.2298 | 1.2991 |
| Two-step | RMSE(m/s) | 0.7234 | 0.7974 | 0.8235 |
| | MAPE (%) | 9.5018 | 10.3757 | 10.5942 |
| | MAE(m/s) | 0.5377 | 0.6079 | 0.6496 |
| | MASE | 1.2119 | 1.3585 | 1.4163 |
| Three-step | RMSE(m/s) | 0.8026 | 0.8685 | 0.8895 |
| | MAPE (%) | 10.4591 | 11.0179 | 11.5248 |
| | MAE(m/s) | 0.7187 | 0.7645 | 0.7983 |
| | MASE | 1.5159 | 1.5748 | 1.6150 |

**Table 10.** Statistical indices obtained by forecasting models for data set B.

| Horizon | Index | Proposal | EEMD-GA-BPNN | EMD-NN |
|---------|-------|----------|--------------|--------|
| One-step | RMSE(m/s) | 0.4456 | 0.4911 | 0.5235 |
| | MAPE (%) | 7.5988 | 8.3497 | 8.6911 |
| | MAE(m/s) | 0.3434 | 0.4131 | 0.4404 |
| | MASE | 0.8786 | 0.9298 | 0.9426 |
| Two-step | RMSE(m/s) | 0.5018 | 0.5465 | 0.5592 |
| | MAPE (%) | 7.9787 | 8.9947 | 9.5173 |
| | MAE(m/s) | 0.3668 | 0.4383 | 0.4690 |
| | MASE | 0.9385 | 0.9886 | 1.1003 |
| Three-step | RMSE(m/s) | 0.5427 | 0.5928 | 0.6270 |
| | MAPE (%) | 8.8626 | 9.8938 | 10.6943 |
| | MAE(m/s) | 0.4178 | 0.4795 | 0.4993 |
| | MASE | 0.9947 | 1.1167 | 1.2975 |

## 6. Conclusion

Considering the signal decomposition, feature selection and parameter optimization, a new compound DAWNN model tuned by HBSA combined with two-stage decomposition EEMD/VMD is proposed for short-term WSF. The two-stage decomposition technique is exploited to address the irregularity of IMF1 using a combination of EEMD and VMD, and this strategy can enhance forecasting performance. A HBSA algorithm integrating BBSA and RBSA was employed to make the feature selection and parameter optimization, simultaneously. BBSA algorithm is exploited as feature selection to eliminate the illusive components in the input matrix determined by PACF method, and RBSA algorithm is applied to tune the input and output weight, and the weighted coefficient. To improve the regression performance of WNN, the integrations of Morlet function and Mexican hat function by weighted coefficient are employed as activation function for the forecasting engine. The comparisons of the proposed model with other individual models and hybrid models mentioned in this study highlight the effectiveness the EEMD/VMD-HBSA-DAWNN model when applied in the WSF. Thus, the proposed EEMD/VMD-HBSA-DAWNN model is an effective WSF strategy.

**Author Contributions:** S.S. designed the principles of the overall work, realized MATLAB-based simulation, and prepared the initial draft of the paper. L.W. proposed some technical comments and the basic idea. J.X. edited some initial draft of the paper. Z.J. investigated the paper.

**Funding:** This work was supported by the Open Research Fund of Wanjiang Collaborative Innovation Center for High-end Manufacturing Equipment, Anhui Polytechnic University under grant GCKJ2018010. The Natural Science Foundation of Anhui Province under grant 1608085MF146, the Natural Science Research Program of Colleges and Universities of Anhui Province under grant KJ2016A062, and the Foundation for talented young people of Anhui Polytechnic University under grant 2016BJRC008. Youth Foundation of Anhui Polytechnic University under grant 2017YQ04.

**Conflicts of Interest:** The authors declare no conflict of interest.

## Appendix A  List of Abbreviations

| | |
|---|---|
| AI | Artificial intelligent |
| ARMA | Autoregressive moving average |
| ARIMA | Autoregressive integrated moving average |
| BBSA | Binary-valued backtracking search algorithm |
| BPNN | Back-propagation neural network |
| BSA | Backtracking search algorithm |
| CEEMDAN | Complementary ensemble empirical mode decomposition with adaptive noise |
| CSA | Coupled simulated annealing |
| DAWNN | WNN with double activations through weighted coefficient |
| EEMD | Ensemble empirical mode decomposition |
| ELMNN | Elman neural network |
| EWT | Empirical wavelet transforms |
| HBSA | Hybrid backtracking search algorithm |
| HGSA | Hybrid gravitational search algorithm |
| IMF | Intrinsic mode function |
| KF | Kalman filter |
| MAE | Mean absolute error |
| MAPE | Mean absolute percent error |
| MASE | Mean absolute scale error |
| MLP | Multilayer perceptron |
| PACF | Partial autocorrelation function |
| RBFNN | Radial basis function neural network |
| RBSA | Real-valued backtracking search optimization algorithm |
| Res | Residual |
| RMSE | Root mean square error |
| SSA | Singular spectrum analysis |
| VMD | Variational mode decomposition |
| WNN | Wavelet neural network |
| WPD | Wavelet packet decomposition |
| WSF | Wind speed forecasting |
| WT | Wavelet Transform |

## Appendix B

**Table A1.** The main parameters used in algorithms.

| Variables in WNN | Values | Variables in WNN | Values |
|---|---|---|---|
| *Input weighted coefficient* $\omega_{i,j}$ | $[-1, 1]$ | *Scale factor* $a_{i,j}$ | $[0.5, 2]$ |
| *Output weighted coefficient* $\omega_{j,k}$ | $[-1, 1]$ | *Position factor* $b_{i,j}$ | $[-3, 3]$ |
| *Weighted coefficient* $\mu$ | $[0, 1]$ | | |
| **Variables in HBSA** | **Values** | **Variables in HBSA** | **Values** |
| *Iteration number T* | 100 | *Dimension number D* | 218 |
| *Population size* (*N*) | 30 | *Parameter F* | *3\*rndn* |

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
