# Peer review of "A New Wind Speed Forecasting Modeling Strategy Using Two-Stage Decomposition, Feature Selection and DAWNN"

_energies, doi:10.3390/en12030334_

Round 1

Reviewer 1 Report

The paper proposes a novel forecasting strategy for multi-step wind speed forecasting (WSF) and illustrates its effectiveness. The proposed forecasting strategy is validated by two sets of real-world wind speed time series that are measured and collected from a wind farm situated in the Anhui, China. Forecasting results, illustrate that the proposed forecasting strategy is an effective and efficient model when employed in WSF regardless of one-, two- or three-step prediction.

As a general comment, I think the paper makes an interesting contribution to the literature by suggesting this new forecasting method. At the same time, I think the paper needs some minor improvements before to be published in this journal.

Broad comments

1)      In the introduction, concerning the issue of renewable energy, I would suggest the authors to add at least a paragraph where the topic of the sustainable transition is discussed. I also suggest to add the following citations:

a.       Linda Ponta, Marco Raberto, Andrea Teglio, Silvano Cincotti, “An Agent-based Stock-flow Consistent Model of the Sustainable Transition in the Energy Sector”, Ecological Economics, Volume 145, Pages 274-300, 2018

b.       Allegra De Filippo, Michele Lombardi, Michela Milano, “User-Aware Electricity Price Optimization for the Competitive Market”, Energies, Volume 10, Pages 1378; 2017

2)      I would suggest the author to cite the following papers where different forecasting technique are compared

a.       Silvano Cincotti, Giulia Gallo, Linda Ponta, Marco Raberto, “Modelling and forecasting of electricity spot-prices: Computational intelligence vs classical econometrics”, AI Communications, Volume 27, Issue 3, Pages 301-314, 2014

3)      In order to let the paper more readable, I would suggest the authors to add a table where the main variables used in algorithms are summarized

4)      Line 299-302: “Two sets of 700 successive wind speed time series are selected randomly from the wind speed collection of 2015. In the empirical samples, as shown in Fig. 4, the foregoing 600 sampling points are utilized to train the HBSA-DAWNN model and the subsequent 100 wind speed time series are employed to test the proposed model”. How these values have been chosen? Do the results change if these values are changed? Please explain.

5)      In order to let the paper more readable, it can be useful to add a table with the main statistical properties of the data used in the forecasting application.

Specific comments

Line 240: please change "time point" with "time step"

Figure 1:  What do “i” and “j” mean? It is not clear. Please modify.

Author Response

Dear Reviewers:

Thank you for your comments concerning our manuscript. Those comments and suggestions are all valuable and very helpful for revising and improving our paper, as well as the important guiding significance to our researches. We have studied comments and made detailed correction according to your’ suggestions carefully. The detailed response to your comments are illustrated in the enclosure. Thank you very much.

Best regards!

Sizhou Sun.

2018.12.1

Reviewer 2 Report

THe contribution is timely and appropriate for the journal. Speed wind forecasting is an important topic in more real world applications.

The authors propose a new method for speed winf forecasting. The results appear correct and the method is solid.

I suggest to include the two following references in order to integrate the state of the art and to introduce in a wider manner the problem.

2014 International Conference on Fractional Differentiation and Its Applications, ICFDA 201425 November 2014, Article number 69674502014 International Conference on Fractional Differentiation and Its Applications, ICFDA 2014; Catania; Italy; 23 June 2014 through 25 June 2014; Category numberCFP1492V-ART; Code 109435

Fractal order evidences in wind speed time series(Conference Paper)

Fortuna, L.aEmail Author,

Nunnari, S.aEmail Author,

Guariso, G.bEmail Author

nternational Journal of Renewable Energy ResearchOpen AccessVolume 6, Issue 3, 2016, Pages 1137-1145

One day ahead prediction of wind speed class by statistical models(Article)

Fortuna, L.a,

Guariso, G.b,

Nunnari, S.aEmail Author

Moreover the paper is well written.

Author Response

Dear Reviewers:

Thank you for your comments concerning our manuscript. Those comments and suggestions are all valuable and very helpful for revising and improving our paper, as well as the important guiding significance to our researches. We have studied comments and made detailed correction according to your’ suggestions carefully. Particularly, we have made the main revisions as below:

1)       English grammar, spelling, expressions, format and other typos are carefully revised.

2)       The parts are modified and marked with red color in the paper.

Comment1: The contribution is timely and appropriate for the journal. Wind speed forecasting is an important topic in more real-world applications. The authors propose a new method for wind speed forecasting. The results appear correct and the method is solid. I suggest to include the two following references in order to integrate the state of the art and to introduce in a wider manner the problem.

a: Fractal order evidences in wind speed time series (Conference Paper)

b: One day ahead prediction of wind speed class by statistical models (Article)

Response1: Thanks for your comments. The two references a and b are very helpful for the paper and                cited in Line 44 and 54, respectively.

Thank you very much.

Best regards!

Sizhou Sun.

2018.12.1

Reviewer 3 Report

Even if the approach is not completely new, in fact there are already a number of papers in the literature in the field of renewable energy that use similar approaches, I think that the paper  can gives some contribution to the literature.

Author Response

Dear Reviewers:

Thank you for your positive comments concerning our manuscript. We have studied comments and made detailed correction according to reviewers’ suggestions carefully. Particularly, we have made the main revisions as below:

1)       English grammar, spelling, expressions, format and other typos are carefully revised.

2)       The parts are modified and marked with red color in the paper.

Thank you very much.

Best regards!

Sizhou Sun.

2018.12.1

Reviewer 4 Report

I have the following comments on the paper:

1) a list of acronyms is mandatory, due to the great number of acronyms used in the paper

2) the strategy should be presented before the methodological part, in order to provide a clear match between the outputs of a previous step of the procedure and the inputs of the next step

2) the majority of the symbols used in the equations of the paper are not defined, thus the entire methodological section lacks a rigorous presentation

3) the definition of MAPE (eq 24) and the definition of MASE (equation 25) are both wrong. In particular, there is the normalizing value lacks in the MAPE equation, and the MASE is not computed with respect to a naive method

4) validating a proposal on two 100-samples datasets of wind speed (for 25 total hours each) is not acceptable and could lead to biased consideration. Please provide numerical experiments for a larger test set (one or more months)

5) which is the computational time required to run the entire forecasting procedure, to build a single prediction?

6) explainations on how the PACF procedure is applied (in particular, some notes on which significance level is selected and which is the critical region are required)

7) how is the optimal number of neurons selected? the RMSE plotted in Fig. 7 refers to which data interval?

8) wind speeds plotted in figs 8a, 8c, 8e, 9a, 9c, 9e are barely readable

9) there is no explaination on the structure (data used, training/testing division,...) for the case 2

10) many hyper-parameters of the proposal are just placed

Author Response

Dear Reviewer:

     Thank you for your comments concerning our manuscript. Those comments and suggestions are all valuable and very helpful for revising and improving our paper, as well as the important guiding significance to our researches. We have studied comments and made detailed correction according to your’ suggestions carefully. 

Kind regards,    Sizhou Sun, 2018.12.24

Round 2

Reviewer 1 Report

 The paper has been improved following the referees’ suggestions, and now, according to me, it is ready to be published in this journal.

Reviewer 2 Report

The paper is very good,

Reviewer 4 Report

There are still some points of concern in the paper.

1) Numerical experiments based on two datasets, each made of 100 samples, are not significant and could lead to biased results. Assessing the validity of a forecasting model requires larger datasets. The authors stated that "The one or more months wind speed forecasting, medium term wind speed prediction, is the subsequent research objective." This is not what it was intended when the authors were asked to "provide numerical experiment for a larger test set". You can run short-term forecast and test your model for one month, two months,..., several years. Just move the forecast origin and you will still develop short-term forecasts, as reported in fig. 6 of the paper.

Testing the method on larger datasets (one or more months) is mandatory in order to assess the validity of the proposal.

2) The authos should better explain what do they mean when they say "For different non-linear samples, there are no uniform standard for selecting the optimal number of hidden neurons. In the study, the optimal number of neurons is determined by experiments."
The first part of their statement is also partially wrong. There is a large theory on the selection selection of model hyper-parameters of forecasting systems. There exist some well-developed and well-structured techniques, such as reserving a portion of the dataser for validation, or making cross-validation in the training set. Since there is no mention of them in the paper, and the only dataset splitting consist in training + testing, I have to assume that you performed no validation.

In such case, the optimal number of neurons can be determined "by experiments", but this should be made only out of sample (i.e., for the test dataset only, 600 samples). Note that this practice is not advisable, since it often leads to overfitting, in particular when there are many hyper-parameters in the developed procedure.
However, tha major problem is that, since the authors stated that the RMSE plotted in Fig 7 refers to the original wind speed data A (which is made of 600+100 samples), it appears that they picked the optimal number of neurons from this analysis. This is "watching the future", which is quite like cheating on forecasts.

I suggest the authors to take more time to review these points, in order to better present their procedure and the results

Author Response

Dear reviewer.

    Thanks for your comments very much. We carefully modify the paper according to your comments. The detailed responses are displayed in the file. Happy new year.

Kind regards,

 Sizhou Sun. 2019.1.4

Round 3

Reviewer 4 Report

I appreciate the authors' efforts and the paper has been significantly improved. I have no further comments on it. Good work.